# SINGLE MOTION DIFFUSION

**Sigal Raab**[*], **Inbal Leibovitch**[*], **Guy Tevet, Moab Arar,**
**Amit H. Bermano and Daniel Cohen-Or**
Tel Aviv University, Israel
{sigal.raab,inbal.leibovitch}@gmail.com

## ABSTRACT

Synthesizing realistic animations of humans, animals, and even imaginary creatures, has long been a goal for artists and computer graphics professionals. Compared to the imaging domain, which is rich with large available datasets, the number of data instances for the motion domain is limited, particularly for the animation of animals and exotic creatures (*e.g*., dragons), which have unique skeletons and motion patterns. In this work, we introduce SinMDM, a Single Motion Diffusion Model. It is designed to learn the internal motifs of a single motion sequence with arbitrary topology and synthesize a variety of motions of arbitrary length that remain faithful to the learned motifs. We harness the power of diffusion models and present a denoising network explicitly designed for the task of learning from a single input motion. SinMDM is crafted as a lightweight architecture, which avoids overfitting by using a shallow network with local attention layers that narrow the receptive field and encourage motion diversity. Our work applies to multiple contexts, including spatial and temporal in-betweening, motion expansion, style transfer, and crowd animation. Our results show that SinMDM outperforms existing methods both qualitatively and quantitatively. Moreover, while prior network-based approaches require additional training for different applications, SinMDM supports these applications during inference. Our project page, which includes links to the code and trained models, is accessible at https://sinmdm.github.io/SinMDM-page.

## 1 INTRODUCTION

Animation of 3D characters is a long pursued task in computer graphics with many applications, from the big screen to virtual reality headsets. It is notoriously known as a time-consuming task done by expert artists. In recent years, neural models have offered faster and less expensive tools for modeling motion (Holden et al., 2016; Petrovich et al., 2022; Raab et al., 2023). In particular, the very recent adaptation of diffusion models into the motion domain provides unprecedented results in both quality and diversity (Tevet et al., 2023; Kim et al., 2022).

These data-driven methods typically require large amounts of data for training. However, motion data is quite scarce. Moreover, for non-human skeletons, it barely exists. The few available datasets contain humanoids only, with fixed topology and bone ratio. Animators often customize a skeleton per character (human, animal, or magical creature), for which common data-driven techniques are irrelevant.

In this work, we present a Single Motion Diffusion Model, dubbed SinMDM, that trains on a *single motion* input sequence. Our model enables modeling motions of arbitrary skeletal topology, which often have no more than one animation sequence to learn from. An *arbitrary topology* refers to the agnostic nature of the architecture towards skeletal structure. Each motion sequence (of any topology) requires its own trained model. SinMDM can synthesize a variety of variable-length motion sequences that retain the core motion elements of the input and can handle complex and non-trivial skeletons. For example, our model can generate a diverse clan based on one flying dragon or one hopping ostrich.

In the motion domain, learning from a single instance has been explored by Li et al. (2022) using GANs (Goodfellow et al., 2014) and by Li et al. (2023) using Motion-graphs (Kovar et al., 2002). We advocate diffusion models (Ho et al., 2020) for single input learning, as gradual denoising enhances descriptive capability and avoids mode-collapse using a rather lightweight scheme that, compared to the prior art, generates high-quality motions. Furthermore, we demonstrate that diffusion models can be effectively utilized with limited data, challenging the notion that they solely rely on large datasets.

To learn local motion sequences, the receptive field must be small enough, analogously to the use of patch-based discriminators (Isola et al., 2017; Li & Wand, 2016) in GAN-based techniques. The use of a narrow receptive field (Fig. 2) promotes diversity and reduces overfitting. We show the importance of narrow receptive fields in our ablation studies.

Most motion diffusion models use transformers (Vaswani et al., 2017). However, vanilla transformers are not suitable for learning a single sequence, as their receptive field encompasses the entire motion.

---

[*]equal contribution.

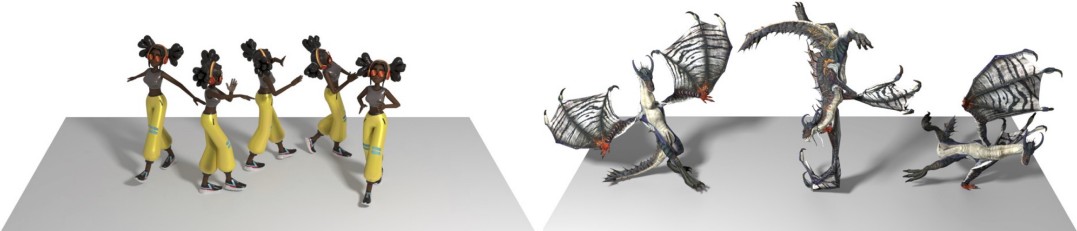

Figure 1: SinMDM learns the internal motion motifs from a single motion sequence with arbitrary topology and synthesizes motions that are faithful to the learned core motifs of the input sequence. **Left:** a girl exercising while walking. **Right:** a breakdancing dragon. Left to right: breakdance uprock, breakdance freeze, and breakdance flair.

A similar challenge arises with UNets (Ronneberger et al., 2015), commonly used in image diffusion models. Its depth, along with global attention layers, creates a receptive field covering the whole motion.

SinMDM leverages the concept of narrow receptive fields and introduces a motion architecture specifically designed with this concept. It combines a shallow UNet model adapted for motion with a QnA (Arar et al., 2022) local attention mechanism, instead of global attention. As a result, SinMDM outperforms prior art both quantitatively and qualitatively, and demonstrates high efficiency with shorter training time and less memory consumption. Imputed to its lightweight architecture, SinMDM can be trained on a single mid-range GPU.

We present many use cases of SinMDM. While neural-network-based prior arts require designated training per application, ours are applied at inference time, with no need to re-train. Moreover, applications that require different dedicated algorithms in prior art, are here grouped together as special cases of the same technique, significantly simplifying their use. One of the applications we present is *Motion Composition*, where a given motion sequence is composed jointly with a synthesized one, either temporally or spatially. Its special cases include in-betweening and motion expansion. Another application that we present is *Harmonization*, along with its special case, style transfer. Here, a reference motion is modified to match the learned motion motifs. It should be emphasized that implementing style transfer using a denoising model is a non-trivial task, and enabling it through motion harmonization is unique. We further present two more applications: *long sequence generation* and *crowd animation*.

In our work, we suggest two comprehensive benchmarks for single-motion evaluation. The first is built upon the artistically crafted MIXAMO (2021) dataset, utilizing metrics that do not require an additional feature-extracting model. The second is based on the HumanML3D (2022) dataset and enables metrics that use latent features, such as SiFID (Shaham et al., 2019). We show that our model outperforms current works on both benchmarks.

Works with limited fidelity or diversity may excel in one metric, but struggle in others. Thus, rather than favoring models that shine on a specific criterion, we use the Harmonic Mean metric, which balances given scores and quantifies their weighted performance.

## 2 RELATED WORK

**Single-Instance Learning** The goal of single-instance generation is to learn an unconditional generative model from a single instance, capturing patch-level statistics for generating diverse content. The instance type varies by input domain, with a majority in imaging. The first works on this topic are SinGAN (Shaham et al., 2019) and InGAN (Shocher et al., 2019). SinGAN uses a patch-based discriminator (Isola et al., 2017; Li & Wand, 2016) and an image pyramid to hierarchically generate diverse results. InGAN uses a conditional GAN to solve the same problem using geometry transformation. More recent approaches include ExSinGAN (Zhang et al., 2021c), which trains multiple modular GANs to model the distribution of structure, semantics, and texture, and ConSinGAN (Hinz et al., 2021), which trains several stages sequentially and improves SinGAN. Many works in the imaging domain follow and improve these works (Asano et al., 2020; Granot et al., 2022; Chen et al., 2021; Lin et al., 2020; Sun & Liu, 2020; Sushko et al., 2021; Yoo & Chen, 2021; Zhang et al., 2022b; Zheng et al., 2021).

Similar to us, Wang et al. (2022) and Nikankin et al. (2023) avoid the pyramid structure and use a UNet with limited depth. This is not directly applicable to the motion domain. Unlike images with a regularized 2D spatial structure, motions consist of non-regularized skeletal joints with a temporal axis and fewer degrees of freedom. Kulikov et al. (2023) construct a multi-scale diffusion process from down-sampled versions of the training image and their blurry versions.

Several works have been done in other domains, e.g., shapes (Wu & Zheng, 2022) and 3D scenes (Son et al., 2023). In the motion domain, Ganimator (Li et al., 2022) follows SinGAN and uses a GAN architecture, with a patch-based discriminator and a temporal pyramid. GenMM (Li et al., 2023) follows

GPNN (Granot et al., 2022), and employs non-parametric patch nearest-neighbor methods, implemented with motion-graphs (Kovar et al., 2002), and yielding higher-quality outputs with a significantly reduced generation time. Yet, nearest-neighbor methods have limited generalization capabilities, and hence are mainly suitable for tasks involving "copying" parts of the input. In this respect, learning-based methods, like Ganimator, offer more applicability as shown in their style-transfer task.

**Diffusion Models**   Diffusion models have been adapted from thermodynamics (Sohl-Dickstein et al., 2015; Song & Ermon, 2020) to the imaging domain(Ho et al., 2020; Song et al., 2020a). A generated image can be further controlled by classifier (Dhariwal & Nichol, 2021) or classifier-free (Ho & Salimans, 2022) guidance. Local editing of images may be viewed as an inpainting problem, in which a portion of the image is held constant while the model denoises the remaining part (Song et al., 2020b; Saharia et al., 2022). In our work, we adapt this technique for spatial and temporal motion composition.

In the motion domain, several works (Zhang et al., 2022a; Kim et al., 2022) introduce diffusion-based synthesis. The prominent one is MDM (Tevet et al., 2023), which utilizes a lightweight network, uses a transformer rather than the common UNet and predicts motion rather than noise. Like MDM, SinMDM presents a lightweight architecture and predicts motion rather than noise. Unlike MDM, our work uses a QnA-based UNet, as the receptive field of a transformer is the full motion, inducing over-fitting.

**Motion Synthesis Models**   Most of motion synthesis models focus on specific tasks, conditioned on some limiting factors, which can be high-level guidance such as action (Petrovich et al., 2021; Guo et al., 2020; Cervantes et al., 2022) or text (Tevet et al., 2022; Zhang et al., 2021a; Petrovich et al., 2022; Ahuja & Morency, 2019; Guo et al., 2022; Bhattacharya et al., 2021; Tevet et al., 2023), can be parts of a motion such as motion prefix  (Aksan et al., 2019; Barsoum et al., 2018; Habibie et al., 2017; Yuan & Kitani, 2020; Zhang et al., 2021b; Hernandez et al., 2019) or in-betweening (Harvey et al., 2020; Duan et al., 2021; Kaufmann et al., 2020; Harvey & Pal, 2018), motion retargeting or style transfer (Holden et al., 2017; Villegas et al., 2018; Aberman et al., 2019; 2020a;b), and even music  (Aristidou et al., 2022; Sun et al., 2020; Li et al., 2021; Lee et al., 2018). Fewer models are fully unconditioned (Holden et al., 2016; Raab et al., 2023; Starke et al., 2022).

The architecture of synthesis models can be roughly divided into recurrent (Habibie et al., 2017; Ghorbani et al., 2020), autoencoder based (Maheshwari et al., 2022; Jang & Lee, 2020), GAN based  (Degardin et al., 2022; Wang et al., 2020; Yan et al., 2019; Yu et al., 2020), normalizing flows based (Henter et al., 2020), and more recently, neural field based (He et al., 2022) and diffusion based (Tevet et al., 2023; Zhang et al., 2022a; Kim et al., 2022; Shafir et al., 2023). Our work belongs to the latter category.

# 3   PRELIMINARY

In this work, we present SinMDM, a novel framework that learns the internal motion motifs of a *single motion* of arbitrary topology and generates a variety of synthesized motions that retain the core motion elements of the input sequence. At the crux of our approach lays a denoising diffusion probabilistic model – DDPM (Ho et al., 2020). See  Appendix A for a recap of DDPM. Our premise is that diffusion models offer generalization without being susceptible to mode collapse.

We present a lightweight model, efficient in time and space and simple in architecture. This is achieved through the gradual denoising process, which enhances the model's descriptive capability. Our generative network is a UNet (Ronneberger et al., 2015) whose global attention layers are replaced by local QnA layers (Arar et al., 2022).

In the rest of this section we describe our motion representation. Next, we describe our method and design choices (Sec. 4), explore various applications enabled by SinMDM (Sec. 5), detail the experiments conducted to validate our approach (Sec. 6), and summarise with conclusions (Sec. 7). The readers are encouraged to watch the supplementary video to get a full impression of our results.

**Motion representation**   A motion sequence is represented by its dynamic and static features, $\mathbf{D}$ and $\mathbf{S}$, respectively. The former differ at each temporal frame (*e.g.*, joint rotation angles), while the latter is temporally fixed (*e.g.*, bone lengths). $\mathbf{D}$ and $\mathbf{S}$ can be combined into global 3D pose sequences using *forward kinematics* (FK). In our research, we focus on synthesizing the *dynamic features*, leaving the static features intact. That is, we predict dynamics for a fixed skeleton topology with fixed bone lengths. For simplicity, we use the term *motion synthesis* for the generation of dynamic features only.

Let $N$ denote the number of frames in a motion sequence, and $F$ denote the length of the features describing a single frame. In the HumanML3D dataset, a frame is redundantly represented with the root position and joint positions, angles, velocities, and foot contact (Guo et al., 2022). For the other datasets used in this work, a frame is represented by joint angles, root positions, and foot contact labels. We represent the dynamic features of a motion by a tensor $\mathbf{D} \in \mathbb{R}^{N \times F}$. Naturally, the convolution for this representation is 1D, convolving on the temporal dimension (of size $N$) and holding $F$ features. More motion representation details can be found in Appendix B.

Figure 2: **Left:** To allow training on a single motion, our denoising network is designed such that its overall receptive field covers only a portion of the input sequence. This effectively allows the network to simultaneously learn from multiple local temporal motion segments. Our denoiser predicts the input sequence from a noisy one. $x_t^0 \ldots x_t^N$ is a motion of $N$ frames at diffusion step $t$. **Right:** Our network is a shallow UNet, enhanced with a QnA local attention layer.

## 4 GENERATIVE NETWORK

Our goal is to construct a model that can generate a variety of synthesized motions that retain the core motion motifs of a single learned input sequence. More formally, we would like to construct the generative network $p_\theta$ (Eq. 2, Appendix A) that synthesizes a motion $\hat{x}_0$ from a noised motion $x_t$.

Unlike traditional single-instance methods that utilize down-sampled instance pyramids for coarse-to-fine learning, our model employs a straightforward architecture without requiring any pyramids.

Our key insight is that internal motifs are learned more effectively with a limited receptive field (Fig. 2 Left). We design SinMDM, a novel generative architecture, accordingly. Our model is a QnA-based degenerated UNet (Fig. 2 Right). UNets (Ronneberger et al., 2015) are commonly used by imaging diffusion models (Nichol et al., 2022). However, training a UNet on a single-input leads to overfitting due to its large receptive field, yielding synthesized sequences that closely resemble the input.

Our first design choice in mitigating this issue is to decrease the depth of the UNet and thereby limit the receptive field width. However, this step alone is not enough, since standard UNets employ global attention layers, resulting in a receptive field that encompasses the entire sequence. A possible solution would be using local attention in non-overlapping windows, like in ViT (Dosovitskiy et al., 2021). Nonetheless, non-interleaving windows tend to limit the cross-window interaction, compromising the model's performance. Our solution is to use QnA (Arar et al., 2022), a state-of-the-art shift-invariant local attention layer, that aggregates the input locally in an overlapping manner, much like convolutions, but with the expressive power of attention. The key idea of QnA is to introduce learned queries, shared by all windows, allowing fast and efficient implementation. Specifically, QnA enables local attention with a temporally narrow receptive field. Our QnA-based UNet is the first to be used in the motion domain, where we plug QnA layers instead of the global attention layers of a vanilla UNet. In this context, a *vanilla UNet* refers to the UNet utilized in early imaging diffusion works (Ho et al., 2020).

QnA is substantially more efficient than global attention in space and time, and our model benefits from this advantage as a byproduct. A detailed description of the QnA layers is available in Appendix D.

In Sec. 6, we validate these design choices. We show the effectiveness of a narrow receptive field, and justify the usage of QnA layers and the choice of a UNet rather than a transformer. In Appendix C, we provide a comprehensive list of the hyperparameters that can be used to reconstruct our results.

## 5 APPLICATIONS

Single-motion learning using diffusion models enables various applications. All our applications are applied at inference time, with no need to re-train. Unlike SinMDM, Ganimator (2022) requires specialized training for most of its applications, and GenMM (2023), is limited to replications hence cannot handle complex applications like harmonization. In the following, we show *Motion Composition* (Sec. 5.1), where a given motion sequence is composed jointly with a synthesized one, either temporally or spatially. Its special cases include *in-betweening*, *motion expansion*, *trajectory control*, and *joints control*. With our *Motion Harmonization* (Sec. 5.2), a reference input motion is altered to align with the learned motion motifs. We illustrate one important special case, *style transfer*. Lastly, we show how straightforward use (Sec. 5.3) of SinMDM enables *one shot long motion generation* and *crowd animation*. The applications presented here are also demonstrated in our supplementary video.

### 5.1 MOTION COMPOSITION

Given an arbitrary reference motion $y$ unseen by our model, and a region of interest (ROI) mask $m$ (Avrahami et al., 2022), our goal is to synthesize a new motion $\hat{x}_0$, such that the regions of interest $\hat{x}_0 \odot m$ are synthesized from random noise, while the complementary area remains as close as possible to the given motion $y$, *i.e.*, $y \odot (1 - m) \approx \hat{x}_0 \odot (1 - m)$, where $\odot$ is element-wise multiplication. The model should output a coherent motion sequence, where the transition between given and synthesized

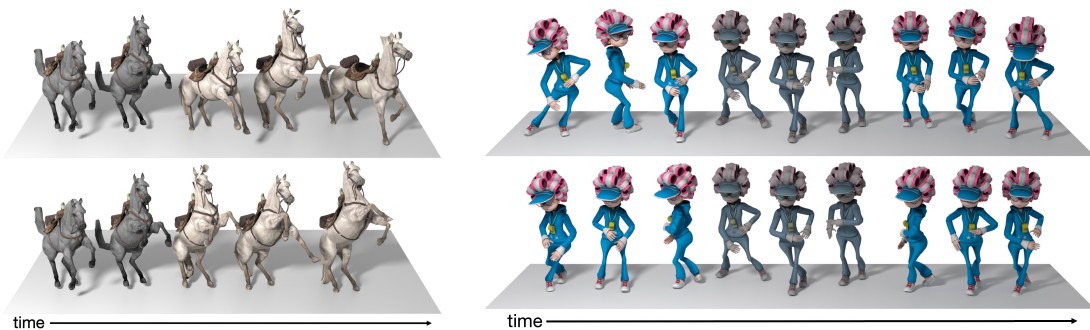

Figure 3: **Temporal composition – motion expansion.** Motion pairs show varied synthesis from a single input. The input, distinguishable by its faded color, is temporally expanded. Note that the parts given as input are identical in both sequences, while the synthesized parts differ. **Left:** synthesize a suffix given a temporal prefix. **Right:** synthesize a prefix and a suffix, given the middle part.

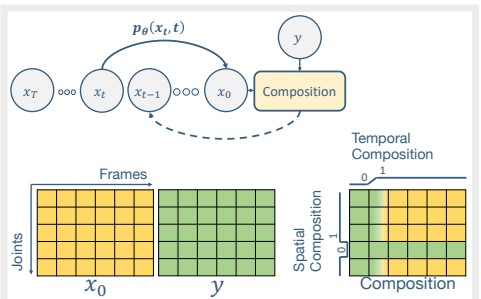

Figure 4: **Motion composition.** Parts from a reference motion $y$, are composed with the synthesized motion $\hat{x}_0$, according to a composition map.

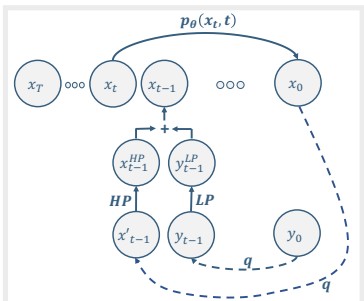

Figure 5: **Harmonization.** During inference, we inject guidance from input $y_0$ by adding its low frequencies $y_{t-1}^{LP}$ in each step $t$.

parts is seamless. Moreover, the reference motion can be an arbitrary one, on which our model has *not* been trained. When the reference motion $y$ is very different from the learned motion, blending between the two becomes less smooth. To mitigate this issue, we change the ROI mask such that the borders between the given and the synthesized motion segments are linearly interpolated, as depicted in Fig. 4. We fix the motion segments that need to remain unchanged and sample the parts that need to be filled in. Each step of the iterative inference process (described in Appendix A) is slightly changed, such that parts of $y$ are assigned into $\hat{x}_0$ according to the indices of the mask. That is, $\hat{x}_0 \odot (1-m) = y \odot (1-m)$.

**Temporal composition – use cases: in-betweening, motion expansion** Temporal composition is the action of filling in selected frame sequences. *In-betweening* (Harvey et al., 2020), depicted in Fig. 6, is a special case of temporal composition, where the filled-in part is at the temporal interior of the sequence, and the reference $y$ is *not necessarily* from the same distribution as the learned motion. Another special case of temporal composition is *motion expansion*, the motion domain's equivalent of image outpainting (Yu et al., 2019; Lin et al., 2021; Teterwak et al., 2019), where the model generates content that resides beyond the edges of a reference motion sequence. In the case of motion expansion, the ROI mask is zeroed in the center frames, and assigned ones in the outer regions. See Fig. 3.

**Spatial composition – use cases: trajectory control, joints control** Motion composition can be applied spatially, by assigning selected joint indices to the ROI mask. In Fig. 7 we illustrate control over the upper body, where the motion of the upper body is determined by a reference motion and assigned to the target motion. The model synthesizes the rest of the joints yielding a motion with the given sequence in the upper body, and with the learned motifs in the lower body. A composition can be both spatial and temporal, and all it takes is an ROI mask where several frame sequences are zeroed, *i.e.*, taken from the reference motion, and in the complementary part, several joints are zeroed (see Fig. 4).

### 5.2 Motion Harmonization

Given a synthesized motion sequence $x_0$, we would like to integrate a portion of an unseen motion, $y$, into it. The portion of $y$ can be either temporal, *i.e.*, several frames, or spatial, *i.e.*, several joints, or both. As visualized in Fig. 5, SinMDM overrides a window in $x_0$ with the desired portion of $y$ and denotes the outcome $y_0$. Next, $y_0$ is harmonized such that it matches the core motion elements learned by our model, using a linear low-pass filter $\phi_N$ as suggested by Choi et al. (2021). Let $x'_{t-1}$ and $y_{t-1}$

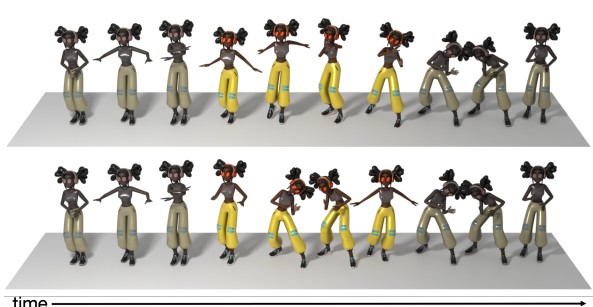

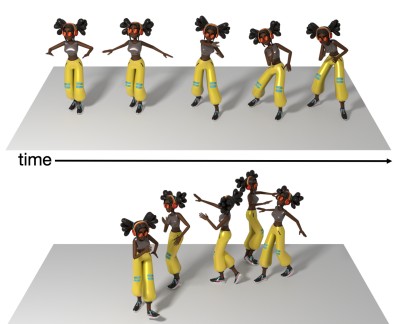

Figure 6: **Temporal composition – In-betweening.** Both rows show results for the same reference input, introducing diverse outputs. The beginning and the end (in faded tones) are given, hence identical in both rows. The center of each motion is synthesized, hence the difference between rows.

Figure 7: **Spatial composition. Top:** "warm-up" – an unseen reference motion. **Bottom:** composed motion. The learned motion is "walk in circle". In the composed result, the top body part performs a warm-up activity, and the bottom walks in a curved path.

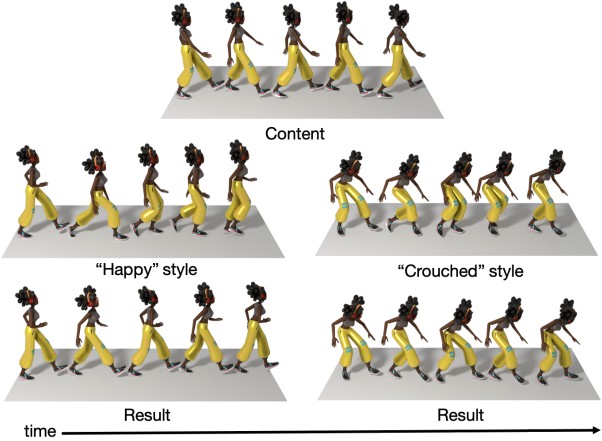

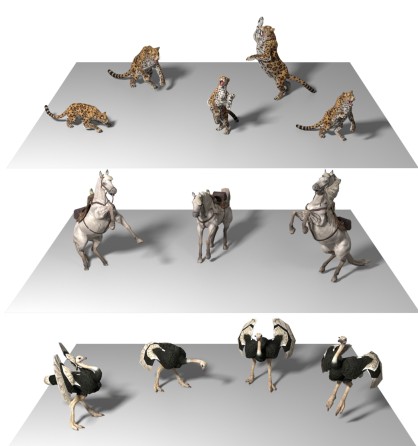

Figure 9: **Style transfer. Top:** one unseen content is applied to both styles. **Left:** a "happy" style learned by the network, with the harmonized result below. **Right:** a "crouched" style. Note the character in both results is using the exact step rhythm and size as in the content motion.

Figure 10: **Crowd animation.** Groups of jaguars, horses, and ostriches. In each group, no motion is like the other, and yet they are all learned from a single motion sequence.

denote the noised version of motions $p_\theta(x_t, t)$ and $y_0$, respectively. The high-frequency details of $x'_{t-1}$ are added to the low-frequency of $y_{t-1}$ via $x_{t-1} = \phi_N(y_{t-1}) + x'_{t-1} - \phi_N(x'_{t-1})$.

Note the difference between harmonization and motion composition: Both assign parts of an unseen sequence $y$ into a synthesized motion $x_0$. However, harmonization changes the assigned part such that it matches the learned distribution, while composition aims to keep it unchanged.

**Style transfer** We implement style transfer (Fig. 9) as a special case of harmonization, where we use all of $y$ instead of a portion of it, hence fully overriding $x_0$. We use a model-learned style motion $x$ and an unseen content motion $y$. After harmonization, the result combines $y$'s content with $x$'s style.

### 5.3 STRAIGHT-FORWARD APPLICATIONS

In this section, we present applications that may require unique methods in prior research but pose no such challenge when employing our model.

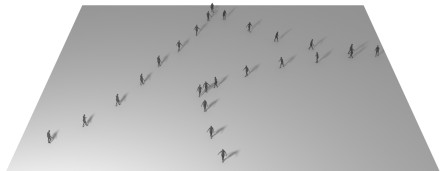

Figure 8: **Long motion.** "a person walking back, turning, and walking back again". Learned on a 10-second sequence and synthesized to 60 seconds.

**Long motion sequences** Our model can synthesize variable-length motions, even very long ones, with no additional training. Imputed to its small receptive field, the model can hallucinate a sequence as long as requested. An example of a one-minute animation is introduced in Fig. 8.

**Crowd animation**  Although trained on a single sequence, during inference SinMDM can generate a diverse crowd, each sampled from a different Gaussian noise $x_T \sim \mathcal{N}(0, I)$, as shown in Fig. 10.

## 6  EXPERIMENTS

Our experiments are held on data from the HumanML3D (2022), Mixamo (2021), and Truebones Zoo (2022) datasets, and on an artist-created animation, using an NVIDIA GeForce RTX 2080 Ti GPU.

### 6.1  BENCHMARKS

We test our framework on two benchmarks, based on the HumanML3D and the Mixamo datasets. These two datasets are different in many aspects. The data in HumanML3D fits the SMPL (Loper et al., 2015) topology, and its users normally use SMPL's mean body definition. In contrast, Mixamo provides 70 characters, each possessing their unique bone lengths and some possessing unique topologies. In addition, the motions in the Mixamo dataset are more diverse and more dynamic.

### 6.2  METRICS

For the Mixamo benchmark, we use the metrics introduced in Ganimator (Li et al., 2022). However, these metrics are based on the values of motion features (*e.g.*, rotation angles) while the usage of deep features is the current best practice (Zhang et al., 2018). Given HumanML3D's capability for deep feature calculation, we utilize it to present our results specifically on these features.

We define a *good* score as being either *high* or *low* depending on the specific metric's preference for higher or lower values. Note that a balance of good scores across all metrics is better than excelling in just a few. In particular, good diversity scores coupled with poor fidelity indicate deviation from the input, while good fidelity scores paired with bad diversity suggest overfitting.

An ideal outcome is a combination of good values for all metrics. For models with mixed scores, a better-scoring model is the one whose scores are more balanced. To this end, we follow established literature (Rijsbergen, 1979; Chinchor, 1992) and suggest the Harmonic Mean metric, which is widely used in Machine Learning for this purpose (Taha & Hanbury, 2015). We compute it as follows: first, we normalize the scores for each metric. Normalization is between zero to the metric's maximum value. If the maximum is not known, we select the 90% percentile of the computed scores. For metrics where lower is better, we subtract the score from the maximum value. Note that a negative value is therefore valid. We compute the Harmonic Mean via $HM = E / \left( \frac{1}{s_1} + \cdots + \frac{1}{s_E} \right)$, where $E$ is the number of metrics in a table and $s_i$ is the normalized score of metric $i$. Additionally, our radar plots (Fig. 11) provide a visual representation of SinMDM's dominance for both benchmarks, considering all metrics.

**Metrics on the Mixamo benchmark**  Our comparison with prior art (Li et al., 2022; 2023) is held on the Mixamo dataset, which is also used by these works. We use metrics suggested by Li et al. (2022).

This group of metrics is applied to the motion itself, and not to deep features. It consists of (a) *coverage*, which is the rate of temporal windows in the input motion $x_0$ that are reproduced by the synthesized one, (b) *global diversity*, measuring the distance between $tess(\hat{x}_0)$ and $x_0$, where $tess(\cdot)$ is a tessellation that minimizes the L2 distance to the input sequence, and (c) *local diversity*, which is the average distance between windows in the synthesized motion $\hat{x}_0$ and their nearest neighbors in the input one.

However, these metrics lack important qualities which we address here: (d) *inter diversity*, the diversity between synthesized motions. We define *intra diversity* to be the diversity between sub-windows internal to a motion and define (e) *intra diversity diff*, which is the difference between the intra diversity of the synthesized motions and that of the input motion.

Additionally, we measure the following time-space efficiency values: (f) number of network parameters, (g) number of iterations, (h) iteration time, and (i) total running time (the product of the last two). A higher score is better for metrics (a)-(d), and a lower one is better for (e)-(i).

**Metrics on the HumanML3D benchmark**  We use this benchmark to measure metrics that are applied on deep features, obtained with a motion encoder by Guo et al. (2022). The computed metrics are (a) *SiFID* (Shaham et al., 2019), which measures the distance between the distribution of sub-windows in the learned motion and a synthesized one, (b) *inter diversity*, which is the LPIPS distance (Zhang et al., 2018) between various motions synthesized out of one input, and (c) *intra diversity diff*, which is the difference between the intra diversity of the synthesized motions and that of the input motion, where intra diversity is the LPIPS distance between sub-windows in one synthesized motion. For metrics (a) and (c) lower is better, and for metric (b) higher is better.

### 6.3  QUANTITATIVE RESULTS

We use the following color scheme in our tables: regular metrics are in grey, the Harmonic Mean metric, which weighs all scores, is in black, and the best scores are in bold.

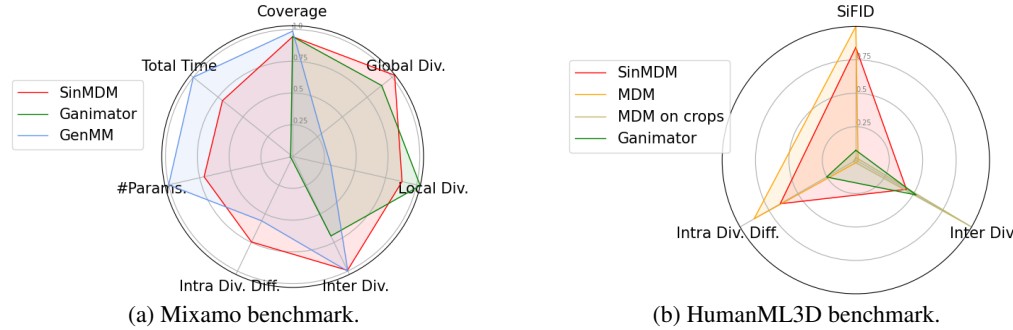

(a) Mixamo benchmark.        (b) HumanML3D benchmark.

Figure 11: **Benchmark results**, depicted as radar plots. Our results present a balanced combination of high scores. The displayed scores were first converted to a 'higher is better' and normalized (Sec. 6.2).

In Tab. 1 and Fig. 11a, we compare SinMDM with prior works and show that it outperforms them. All the metrics are computed separately on each benchmark motion and then averaged. The metrics that measure time were computed on benchmark motion number 9 only.

To evaluate our performance vs. another motion diffusion model, we compare it with two variations of the MDM (Tevet et al., 2023) framework. The first is a vanilla MDM, trained on a single-motion. The second is a variation of MDM in which we extract short crops out of the single-motion sequence and train an MDM on them. Note that the second variation holds a narrow receptive field.

The comparison is conducted on the HumanML3D dataset with metrics based on deep features. The results are shown in Tab. 2 and Fig. 11b. MDM yields complete overfit, thus its SiFID and intra-diversity scores are perfect (indicating similarity to the input motion), but its inter-diversity scores are low. The overfit of MDM is caused by the global attention it uses. On the other hand, the quantitative results for the second MDM variation indicate divergence from the input motion motifs. These quantitative results are supported by the qualitative results in our supplementary video.

Finally, we reinforce our quantitative results in a user study in which users evaluate model superiority based on diversity, fidelity, and quality. In the study, we compare our model vs. other works. Each pair of models is compared over 8 different motions, and each such comparison is judged by 10 distinct users. The results (Fig. 12) show that our model is significantly preferred by the users. Screenshots from our user study are provided in Appendix E.

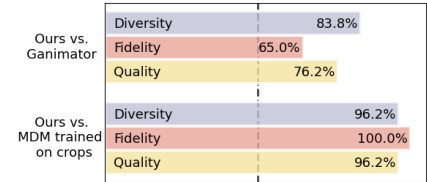

Figure 12: **User study.** The dashed line marks 50%.

### 6.4 QUALITATIVE RESULTS

Our supp video reflects the quality of our results. It presents multiple synthesized motions, as well as a comparison to other works. Additionally, Fig. 13 depicts SinMDM vs. current work. Other works exhibit mode collapse, overfitting, or jittery motion, while SinMDM demonstrates a coherent motion.

### 6.5 ABLATION

We examine architectural variations on the HumanML3D benchmark and present the results in Tab. 3. First, as many motion diffusion models favor a transformer over a UNet (Tevet et al., 2023; Kim et al., 2022), we measure the scores for a QnA-based transformer (row 1). To refrain from overfitting, we apply QnA layers within the transformer as we do with the UNet. In addition, to promote diversity and permit the rearrangement of motion components, we employ relative temporal positional embeddings (Shaw et al., 2018; Press et al., 2022; Su et al., 2021) instead of the existing global ones. However, the QnA-based transformer attains a bad SiFID score, indicating poor fidelity to the input motion.

Table 1: **Mixamo benchmark.** SinMDM demonstrates a significant advantage in the harmonic mean metric. Compared to Ganimator, it leads in all metrics but one. Recall that GenMM requires no training time/space, reflected in zero values at the right-hand side of the table. However, its inability to generalize results in inferior scores for most diversity metrics, leading to a lower harmonic mean.

| | Coverage ↑ | Global Div. ↑ | Local Div. ↑ | Inter Div. ↑ | Intra Div. Diff. ↓ | #Param. (M) ↓ | #Iter. (K) ↓ | Iter. Time (s) ↓ | Tot. Time (h) ↓ | Harmon. Mean ↑ |
|---|---|---|---|---|---|---|---|---|---|---|
| Ganimator | 94.3 | 1.24 | 1.17 | 0.09 | 0.13 | 21.7 | 60 (15×4) | 0.36 | 6.0 | -0.51 |
| GenMM | 98.7 | 0.4 | 0.35 | 0.13 | 0.05 | 0 | 0 | 0 | 0 | 0.56 |
| SinMDM (Ours) | 94.3 | 1.42 | 1.00 | 0.13 | 0.03 | 5.26 | 60 | 0.09 | 1.5 | **0.84** |

Table 2: **HumanML3D benchmark**. Comparison with two variants of the diffusion model MDM. The first exhibits overfitting and the second lacks fidelity to input. Our model attains good and balanced scores in all metrics.

|  | SiFID↓ | Inter Div. ↑ | Intra Div. Diff.↓ | Harmon. Mean ↑ |
|---|---|---|---|---|
| MDM (2023) | 0.01 | 0.03 | 0.14 | 0.05 |
| MDM on crops | 13.94 | 1.64 | 1.83 | -0.46 |
| Ganimator | 11.01 | 0.87 | 0.86 | 0.15 |
| SinMDM (Ours) | 1.87 | 0.73 | 0.40 | **0.60** |

Table 3: **Ablation** on the HumanML3D benchmarks. Row 1: QnA-based transformer. Rows 2,3: comparing receptive field widths. Row 1 exhibits divergence and Row 2 indicates overfit. Abbr.: *r.f.* → receptive field, $d$ → depth.

|  | SiFID ↓ | Inter Div. ↑ | Intra Div. Diff. ↓ | Harmon. Mean ↑ |
|---|---|---|---|---|
| Transformer w/ QnA | 5.99 | 1.74 | 0.57 | 0.47 |
| UNet w/ QnA |  |  |  |  |
| wide r.f. (d=3) | 0.69 | 0.20 | 0.34 | 0.28 |
| narrow r.f. (d=1) (Ours) | 1.87 | 0.73 | 0.40 | **0.59** |

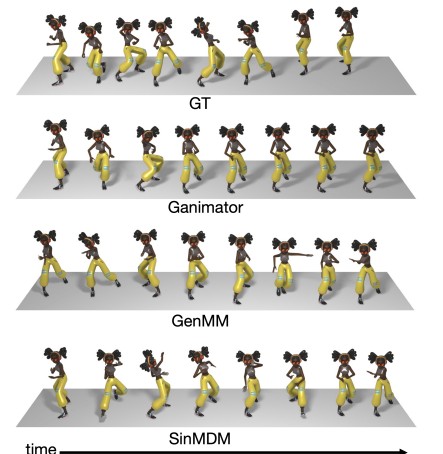

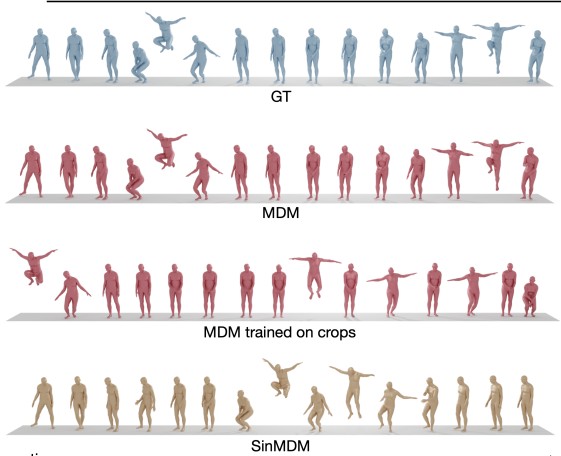

(a) **Mixamo dataset.** Motion *punch to elbow*. SinMDM outperforms Ganimator, which experiences mode collapse, and is comparable in quality to the patch nearest-neighbor method GenMM.

(b) **HumanML3D dataset.** MDM exhibits overfit, and MDM trained on crops exhibits jittery motion, *e.g.*, when transferring from standing to jumping without bending the knees before and after.

Figure 13: **Qualitative comparison.**

We continue by confirming that a narrow receptive field produces good results while a wider one induces overfit (rows 2,3). In order to do so, we examine a fixed architecture (QnA-based UNet) with two receptive field widths. We control the width by tweaking the depth of the UNet. Indeed we observe that the model with the wide receptive field overfits (replicates) the input motion, as its inter-diversity is bad while its SiFID and intra-diversity are good.

Note that due to the mixed scores (that indicate either overfit or divergence), the usage of the Harmonic Mean metric is essential as it allows for the assessment of the combined strength of all scores. In addition, in Appendix F we show an ablation study of the smoothness fidelity score.

## 7 CONCLUSIONS

We have explored the use of diffusion models on single motion sequence synthesis and designed a motion denoising transformer with a narrow receptive field. Training on single motions is particularly useful in motion domains, where the number of data instances is scarce. Particularly, for animals and imaginary creatures, which have unique skeletons and distinctive motion motifs. The motion of such creatures cannot be captured easily nor learned from the human motion data available.

Our experiments across datasets show that our lightweight diffusion-based method significantly outperforms current work both quantitatively and qualitatively. Moreover, our approach allows the synthesis of particularly long motions and enables a variety of motion manipulation tasks, including spatial and temporal in-betweening, motion expansion, harmonization, style transfer, and crowd animation.

The innate limitation of our method, common to single-instance models in all domains, is the limited ability to synthesize out-of-distribution. Another limitation, also common to all single-instance models, is the inability to set generated sub-motions in a specific order, when such order matters (e.g., certain dance moves). This can be addressed by enlarging the receptive field (at the cost of lower diversity). The main limitation of our approach, due to the iterative nature of diffusion models, is the relatively long inference time.

Finally, our work shows that diffusion models can learn from limited data, contrary to their reputation for requiring large amounts of data. Nevertheless, in the future, we aim to address the single input limitations, by possibly learning from available motion data of creatures with rather compatible skeletons.

## 8 ACKNOWLEDGMENTS

We are grateful to Panayiotis Charalambous, Andreas Aristidou and Brian Gordon for reviewing earlier versions of the manuscript. This research was supported in part by the Israel Science Foundation (grants no. 2492/20 and 3441/21), Len Blavatnik and the Blavatnik family foundation, and the Tel Aviv University Innovation Laboratories (TILabs).

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

## APPENDIX

This Appendix adds details on top of the ones given in the main paper. While the main paper stands on its own, the details given here may shed more light.

In Appendix A we provide a concise recap of Denoising Diffusion Probabilistic Models (DDPM). Appendix B describes the dynamic features predicted by our network, and C details the hyperparameters used by it. In Appendix D we describe the QNA framework, and in Appendix E we depict screenshots from our user study. Lastly, Appendix F presents a study about motion smoothness, its correlation to receptive field width, a metric to measure it, and ways to ameliorate it.

## A  DENOISING DIFFUSION PROBABILISTIC MODELS (DDPM) – RECAP

DDPMs (Ho et al., 2020) have become the de-facto leading generative networks technique. While they have primarily dominated the imaging domain (Dhariwal & Nichol, 2021), recent works have successfully applied this approach in the motion domain (Tevet et al., 2023; Zhang et al., 2022a). Denoising networks learn to convert unstructured noise to samples from a given distribution, through an iterative process of progressively removing small amounts of Gaussian noise.

Given an input motion sequence $x_0$, we apply a Markov noising process of $T$ steps, $\{x_t\}_{t=0}^T$, such that

$$q(x_t|x_{t-1}) = \mathcal{N}(\sqrt{\alpha_t}x_{t-1}, (1-\alpha_t)I), \tag{1}$$

where $\alpha_t \in (0,1)$ are constant hyper-parameters. When $\alpha_t$ is small enough, we can approximate $x_T \sim \mathcal{N}(0, I)$.

We apply unconditional motion synthesis that models $x_0$ as the reversed diffusion process of gradually cleaning $x_T$, using a generative network $p_\theta$. Following Tevet et al. (2023) we choose to predict the input motion, denoted $\hat{x}_0$ (Ramesh et al., 2022) rather than predicting $\epsilon_t$, hence

$$\hat{x}_0 = p_\theta(x_t, t). \tag{2}$$

We apply the widespread diffusion loss, via

$$\mathcal{L}_{\text{simple}} = \mathbb{E}_{t\sim[1,T]}\|x_0 - p_\theta(x_t,t)\|_2^2. \tag{3}$$

Synthesis at inference time is applied through a series of iterations, starting with pure noise $x_T$. In each iteration, a clean version of the current sample $x_t$ is predicted using a generator $p_\theta$. This predicted clean sample $\hat{x}_0$ is then "re-noised" to create the next sample $x_{t-1}$, with the process being repeated until $t=0$ is reached.

## B  MOTION REPRESENTATION – ADDITIONAL DETAILS

This section completes the Motion Representation section in the main paper.

In this section, we describe the dynamic features predicted by our network.

Recall that $N$ denotes the number of frames in the sequence, and $F$ denotes the length of the features of all joints together in a single motion frame.

Let $J$ denote the number of skeletal joints, and let $Q$ denote the number of rotational features, where rotational features may be Euler angles, quaternions, 6D rotations, etc. Let $C$ denote the number of joints that are prone to contact the ground. Clearly, a human, a spider, and a snake possess different values of $C$.

When using the HumanML3D (Guo et al., 2022) dataset, we adhere to its representation, in which a single pose is defined by

$$p = (\dot{r}^a, \dot{r}^x, \dot{r}^z, r^y, j^p, j^v, j^r, c^f) \in \mathbb{R}^F,$$

where $\dot{r}^a \in \mathbb{R}$ is the root angular velocity along the Y-axis. $\dot{r}^x, \dot{r}^z \in \mathbb{R}$ are root linear velocities on the XZ-plane, and $r^y \in \mathbb{R}$ is the root height. $j^p \in \mathbb{R}^{3J}$, $j^v \in \mathbb{R}^{3J}$ and $j^r \in \mathbb{R}^{6J}$ are the local joint

Table 4: Our choice of hyperparameters, given with the same names as used in the code.

| Name | Value |
|------|-------|
| UNet related | |
| num_channels | 256 |
| channel_mult | 1 |
| num_res_blocks | 1 |
| kernel_size | 3 |
| use_scalse_shift_norm | True |
| use_checkpoint | True |
| use_attention | True |
| use_qna | True |
| QnA related | |
| head_dim | 32 |
| num_heads | 4 |
| Diffusion related | |
| diffusion_steps | 1000 |
| noise_schedule | cosine |
| Training related | |
| batch_size | 64 |
| dropout | 0.5 |
| lr_method | ExponentialLR |
| lr_gamma | 0.99998 |
| num_steps | 60000 |
| padding_mode | zeros |
| warmup_steps | 0 |
| weight_decay | 0 |

positions, velocities, and rotations with respect to the root, and $c^f \in \mathbb{R}^4$ are binary features denoting the foot contact labels for four foot joints (two for each leg).

When using data from other datasets, we adhere to the representation used by Li et al. (2022), so we can conduct a fair comparison with their results. Their representation consists of a 3D root location, a rotation angle for each joint, and foot contact labels. Altogether, for a general representation $D \in \mathbb{R}^{N \times F}$, we have got $F = 3 + JQ + C$.

The rotations in both representations are defined in the coordinate frame of their parent in the kinematic chain, and are represented by the 6D rotation features ($Q = 6$) Zhou et al. (2019), which yields the best result in many works (Qin et al., 2022; Petrovich et al., 2021).

A growing number of works use foot contact labels (Gordon et al., 2022; Raab et al., 2023) to mitigate common foot sliding artifacts. Let $\mathbf{C}$ denote the set of joints that contact the ground in the subject whose motion is being learned such that $C = |\mathbf{C}|$. The foot contact labels are represented by $\mathbf{L} \in \{0, 1\}^{N \times C}$.

When a dataset provides foot contact label information (Guo et al., 2022), we use it as is. When a dataset does not provide them, we calculate it as done by Li et al. (2022), via

$$\forall j \in \mathbf{C}, n \in [1, N]: \qquad \mathbf{L}^{nj} = \mathbf{1}[\|\Delta_n \text{FK}([\mathbf{D}, \mathbf{S}])^{nj}\|_2 < \epsilon], \tag{4}$$

where $\Delta_n \text{FK}([\mathbf{D}, \mathbf{S}])^{nj}$ denotes the velocity of joint $j$ in frame $n$ retrieved by a forward kinematics operator, and $\mathbf{1}[\cdot]$ is an indicator function.

## C  HYPERPARAMETERS AND TRAINING DETAILS

In Tab. 4 we detail the values of the hyperparameters that have been used to produce the results shown in this work. Our models have been trained on an NVIDIA GeForce RTX 2080 Ti GPU.

## D  QNA RECAP

QnA layers (Arar et al., 2022) are a fundamental component in our suggested architecture. In this section, we provide an overview of its underlying implementation and illustrate it in Fig. 14. In particular,

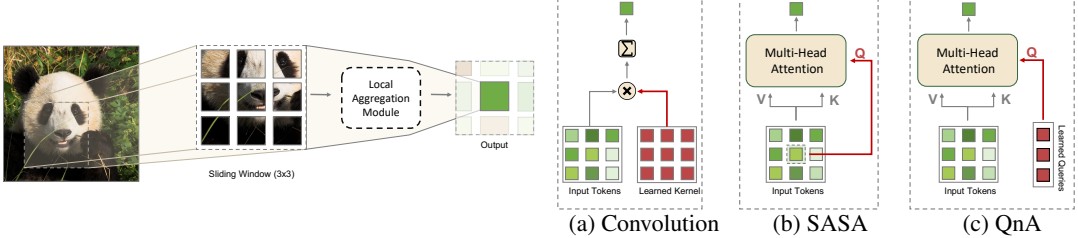

(a) Convolution   (b) SASA   (c) QnA

Figure 14: QnA overview (extracted from the QnA paper). Left: Local layers may utilize various approaches to overlapping windows. (a) Convolutions apply aggregation by learning shared weighted filters. (b) SASA (2019) combines window tokens through self-attention. (c) QnA use shared learned queries across windows, maintaining the expressive power of attention while achieving linear space complexity.

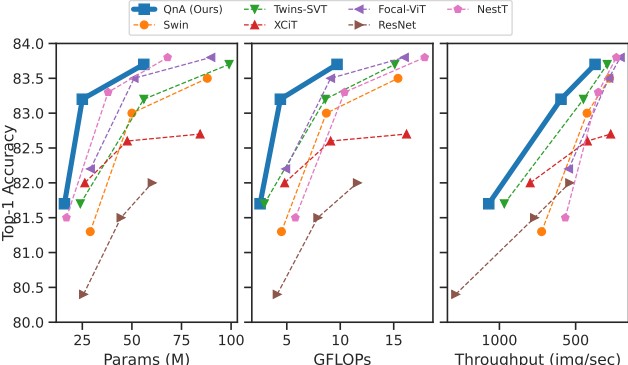

Figure 15: QnA demonstrates better accuracy-efficiency trade-off compared to state-of-the-art baselines (extracted from the QnA paper).

QnA is an efficient attention-based layer, which operates in a shift-invariant manner. For every $k$-size window, the output is calculated using the self-attention mechanism which is commonly used in the transformer architecture (Vaswani et al., 2017). The self-attention is calculated by first projecting the input features into keys $K = XW_K$, values $V = XW_V$, and queries $Q = XW_Q$ via three linear projection matrices $W_K, W_V, W_Q \in \mathbb{R}^{D \times D}$. Then, the output of the self-attention operation is defined by:

$$\begin{aligned}
\mathbf{SA}(X) &= \mathbf{Attention}\left(Q, K\right) \cdot V \\
&= \mathbf{Softmax}\left(QK^T / \sqrt{D}\right) \cdot V.
\end{aligned} \qquad (5)$$

Instead of performing the pricey query-key operation, QnA detours from extracting the queries from the window itself and directly learns them for the whole-training data (see Fig. 14c). Learning the queries preserves the expressive capability of the self-attention mechanism and enables an efficient implementation that relies on simple and fast operations. In particular, a single query $\tilde{q}$ is learned, and the attention is applied locally for every $k$-size window. Therefore, the output at entry $z_i$ becomes:

$$z_i = \mathbf{Attention}\left(\tilde{q}, K_{\mathcal{N}_i}\right) \cdot V_{\mathcal{N}_i}, \qquad (6)$$

where $\mathcal{N}_i$ is the set $k$-neighbourhood of frame $i$.

QnA exhibits state-of-the-art accuracy-efficiency trade-off, as depicted in Fig. 15.

Table 5: **Smoothness error across varying receptive field widths, on the Mixamo benchmark.** Each row shows an architecture with an increasing receptive field width, with the right column displaying the smoothness error. The top row refers to our best-performing architecture (same ones as in Tab. 1).

| | Coverage ↑ | Global Div. ↑ | Local Div. ↑ | Inter Div. ↑ | Intra Div. Diff. ↓ | Smoothness Diff. (× 1e-5) ↓ |
|---|---|---|---|---|---|---|
| no. layers=1 no. res blocks = 1 (baseline) | 94.3 | 1.42 | 1.00 | 0.13 | 0.03 | 9.8 |
| no. layers=1 no. res blocks = 2 | 97.9 | 1.22 | 0.83 | 0.13 | 0.03 | 2.9 |
| no. layers=2 no. res blocks = 1 | 99.4 | 0.99 | 0.68 | 0.12 | 0.04 | 1.9 |

Table 6: **Smoothness error across varying $\lambda_{smooth}$ values, on the Mixamo benchmark.** The left column shows increasing values of $\lambda_{smooth}$, and the right column displays the smoothness error, which decreases as $\lambda_{smooth}$ increases. The top row, where $\lambda_{smooth} = 0$, corresponds to our best-performing architecture to date. Introducing the smoothness loss paves the way for a new best-performing architecture, as using $\lambda_{smooth} = 1$ attains better smoothness with no degradation at the other metrics.

| $\lambda_{smooth}$ | Coverage ↑ | Global Div. ↑ | Local Div. ↑ | Inter Div. ↑ | Intra Div. Diff. ↓ | Smoothness Diff. (× 1e-5) ↓ |
|---|---|---|---|---|---|---|
| 0 (baseline) | 94.3 | 1.42 | 1.00 | 0.13 | 0.03 | 9.8 |
| 0.1 | 94.1 | 1.43 | 1.02 | 0.13 | 0.04 | 7.7 |
| 1 | 94.0 | 1.43 | 1.02 | 0.13 | 0.05 | 4.8 |
| 10 | 92.8 | 1.48 | 1.10 | 0.13 | 0.07 | 2.6 |
| 100 | 89.4 | 1.50 | 1.17 | 0.12 | 0.12 | 1.5 |

# E  USER STUDY – SCREENSHOTS

Our user study displays several video clips on each screen, requesting the user to select the one that is more suitable to the examined attribute, which is either quality, fidelity, or diversity. Screenshots from a representative video for each attribute are shown in Fig. 16.

# F  SMOOTHNESS STUDY

We target our generated motions to closely match the input motion's smoothness. Smoothness varies from motion to motion, for example, a slow walk is characterized by very smooth and flowing motions, while a fast dance often exhibits abrupt transitions. Next, we present a metric to measure the smoothness fidelity. Then, we show that the smoothness error is strongly related to the receptive field width, and lastly, we present a way to improve the smoothness score while keeping our receptive field narrow.

**Metric**  To assess the fidelity of our generated motions to the smoothness of the input motion, we introduce a metric that quantifies the difference between the ground truth smoothness and that of the generated motions. The metric we employ is the acceleration error of each joint, first proposed by Kanazawa et al. (2019) and subsequently adopted by many others (Kocabas et al., 2020; Gordon et al., 2022). The acceleration at joint $j$ is measured using

$$\text{Acc}_j = \mathbb{E}_{n \in [1...N-1]} \|\text{FK}(x)^{(n-1)nj} - 2\text{FK}(x)^{nj} + \text{FK}(x)^{(n+1)nj}\|_2^2, \tag{7}$$

where $\text{FK}(\cdot)^{nj}$ denotes the location of joint $j$ in frame $n$ retrieved using a forward kinematics operator, and $x$ is a motion (given or synthesized).

The acceleration error is measured using

$$\mathcal{L}_{smooth} = \mathbb{E}_{j \in J} \|\mathbb{E}_{x_T \sim \mathcal{N}(0,I)} [\text{Acc}_j(G(x_T))] - \text{Acc}_j(x)\|_2^2, \tag{8}$$

where $G(\cdot)$ is the inference process and $x$ is the input motion on which our model has been trained.

**Varying receptive field study**  In Tab. 5 we compare smoothness scores for varying receptive field widths. Recall, that a UNet layer is composed of a varying number of residual blocks and an attention layer. First we present our baseline, which is the architecture presented in the main paper. Recall that our best-performing model uses a UNet of depth 1, with one residual block within it. Adding one more residual block to a UNet of depth 1 (row #2) widens the receptive field, and results in better smoothness error. Using a UNet of depth 2 results in an even wider receptive field, and a better smoothness error. We conclude that widening the receptive field decreases the smoothness error. Regrettably, expanding the receptive field leads to a significant decline in the quality of the generated motion, as indicated by the diversity metrics, hence it is not a practical way to get better smoothness fidelity.

Next, we suggest a method to enhance smoothness while maintaining our favorable metric scores.

**Loss study**  To get better smoothness without increasing the receptive field, we have used the smoothness term as a loss, denoted $\mathcal{L}_{smooth}$. Accordingly, our loss is changed to

$$\mathcal{L} = \mathcal{L}_{simple} + \lambda_{smooth}\mathcal{L}_{smooth}. \tag{9}$$

Table 6 displays our metric scores on the Mixamo benchmark, for varying values of $\lambda_{smooth}$. The table shows that increasing $\lambda_{smooth}$ decreases the smoothness error. Up to a value of $\lambda_{smooth} = 1$, the other metric scores remain comparable to the original ones, and for larger values of $\lambda_{smooth}$ we see a degradation in the coverage and the intra diversity diff. We conclude that training with a value of $\lambda_{smooth} = 1$ would provide the best results.

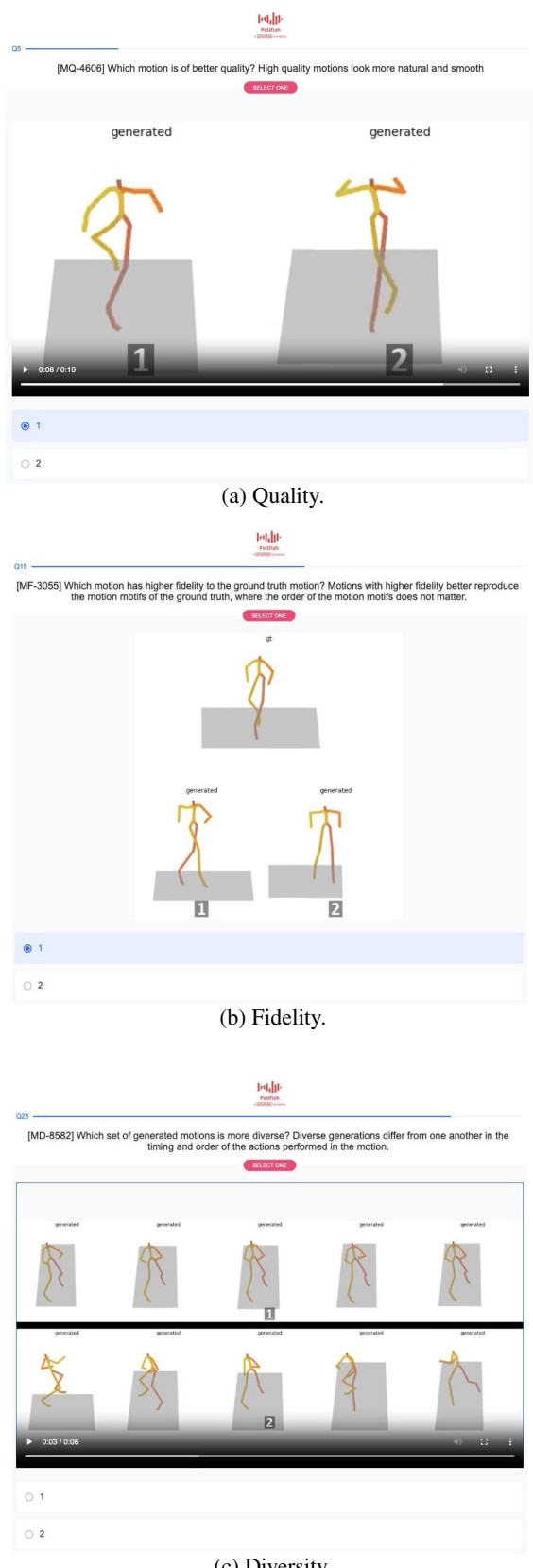

(a) Quality.

(b) Fidelity.

(c) Diversity.

Figure 16: Screenshots from our user study. Note that each human figure in the screenshot is played as a video.

