# OpenReview forum: "Single Motion Diffusion"
_ICLR.cc/2024/Conference — ICLR 2024 spotlight_

### Official Review · Reviewer_BL7j · 2023-10-30

**Soundness:** 4 excellent
**Presentation:** 4 excellent
**Contribution:** 3 good
**Rating:** 8
**Confidence:** 4

**Summary:**

The paper proposes "Single Motion Diffusion" to learn a generative model of skeletal motions from small training data. The key architectural ideas are 1. shallow U-Net and 2. QnA attention model for local receptive fields. The paper demonstrates applications in the temporal composition (temporal in-painting for in-betweening and motion expansion), spatial composition (spatial in-painting by constraining a part of the skeletal tree with an unseen motion), harmonization (mix low-frequency unseen motion during denoising), and style transfer using harmonization. The method is compared numerically against Ganimator [Li et al. 2022] and GenMM [Li et al. 2023] for the coverage (the rate of generated motions reproducing the input), global diversity, local diversity, inter-diversity, intra-diversity difference, and computational efficiency metrics (number of parameters and total running time). The method is also compared against MDM trained on a single clip and MDM trained on a cropped clip to simulate the narrow receptive field regarding SiFID, inter-diversity, and intra-diversity differences. The numerical evaluations are backed up by the user studies.

**Strengths:**

The core takeaway is reducing the temporal receptive field by tweaking existing architectures (reducing the U-Net depth and using the QnA transformer) allows the training of generative motion models with a very small amount of data. While the idea is simple, the technique shows promising results on important animation tasks.

Thorough numerical evaluations and user studies support the claim. The visual quality is also fine but with some caveats (see Weakness).

The technique is reproducible with the provided code, data, and pretrained models.

**Weaknesses:**

The flip side of the simplicity of the technique is that the technical novelty may not be so significant.

It seems like there is a trade-off to reducing the temporal receptive field. The generated results often show abrupt transitions, such as unnaturally fast turns. I believe this is due to the limited receptive field not seeing the full segment of some longer actions. Can authors discuss this? This may indicate that we need more metrics to measure the visual quality of motions.

**Questions:**

(Minor comment)
From the abstract:
> learn the internal motifs of a single motion sequence with arbitrary topology

I initially thought "arbitrary topology" meant cases like generating the dragon motions from the model trained on human skeletons. I now know that this just means that the architecture itself is agnostic to the skeletal structure but the trained model must be applied to the same skeletal structure. I hope there is a way to phrase this better to reduce the confusion.

---

> ### Author Response · Authors · 2023-11-21
> **Reply to reviewer ​​BL7j (R4)**
>
> **“It seems like there is a trade-off to reducing the temporal receptive field. … Can authors discuss this? This may indicate that we need more metrics to measure the visual quality of motions.”**
>
> Following this question, we added Appendix F, presenting a study about motions smoothness. In this study, we describe a metric that measures the smoothness fidelity between the generated motion to the ground truth on which the network has been trained.  We show that using a wider receptive field improves smoothness fidelity but degrades the other metrics, hence reducing motion quality. Finally, we introduce a method to enhance smoothness fidelity while maintaining a narrow receptive field, thereby preserving the quality of the results.
>
> **“I initially thought "arbitrary topology" meant cases like generating the dragon motions from the model trained on human skeletons.…”**
>
>  We added a clarification in the introduction (in blue). Thank you for bringing the need for this clarification to our attention.

---

> > ### Comment · Reviewer_BL7j · 2023-11-21
> >
> > Is eq. 8 in the appendix the smoothness loss L_smooth? Does it make sense to clarify this like eq. 3, the definition of L_simple, in the appendix?
> >
> > Can authors provide videos of the smoothness study?

---

> > > ### Author Response · Authors · 2023-11-22
> > > **Reply to reviewer ​​BL7j (R4)**
> > >
> > > We changed eq. 8 according to your suggestion and uploaded a new pdf. Thank you.
> > >
> > > In addition, we uploaded a new supplementary zip, in which the file smoothness*.mov shows videos related to the smoothness study.
> > >
> > > Description of the smoothness video:
> > >
> > > At the top right of each slide is the input motion.
> > >
> > > In the bottom-right corner of each slide are two generated motions. The one on the right has been trained with the smoothness loss (𝜆𝑠𝑚𝑜𝑜𝑡ℎ=1), while the one on the left has been trained without it.
> > >
> > > Both motions share the same seed, facilitating a temporal similarity in their sub-motions for a more straightforward comparison.
> > > When played at normal speed, the motions provide a general impression, but the details are too fast to discern.
> > >
> > > On the left-hand side of each slide, we showcase a slow-motion sequence spanning several frames, highlighting the enhancement achieved in motions trained with the smoothness loss.
> > >
> > > The first slide displays an unnatural turn executed by the model trained without the smoothness loss. The second slide depicts a higher-frequency, unnatural leg movement in the model trained without the smoothness loss. The leg changes directions too rapidly, initiates a step, and retracts without completing the step. This leg movement does not align with the motifs of the input motion.

---

> > > > ### Comment · Reviewer_BL7j · 2023-11-22
> > > >
> > > > Looks all good to me.
> > > >
> > > > The authors have addressed all of my concerns. I will keep my score as is.

---

### Official Review · Reviewer_gTdv · 2023-10-31

**Soundness:** 3 good
**Presentation:** 3 good
**Contribution:** 3 good
**Rating:** 8
**Confidence:** 3

**Summary:**

Paper introduces a generative diffusion model for motion synthesis. The model is a UNet that operates on motion data with a specialized efficient local attention layer (QnA) in place of global attention. The key intuition of this work is restricting the capacity (receptive field) of the underlying UNet to improve generalization and avoid mode collapse. Results are provided on several tasks, including motion composition, harmonization and generation, and seem to be state-of-the-art.

**Strengths:**

- Paper is well-written and is easy to follow. Supplementary material is of high quality.
- The key technical novelty - to reduce receptive field - is simple and is easy to implement, and makes sense in the limited data regime.
- Qualitatively results look impressive. User study and benchmarks also confirm performance of the method is state-of-the-art.

**Weaknesses:**

- Authors propose a solution that fixes a specialized version of UNet (presumably similar to the one used for image generation) - by introducing local attention mechanism. I understand that it might be a way to repurpose an existing architecture developed for a different task, but it does seem like there are exist a wide range of existing architectures (1d convnets / wavenets), which exhibit the same property by construction. It is a bit unclear why is the vanilla UNet considered in the first place?
- (Novelty, minor): diffusion models have been used for motion synthesis, the QnA has been used before, although in different context.
Misc:
- afaik vanilla UNet [Ronneberger'2015] does not have attention layers, do you imply the unet similar to that used in stable diffusion implementation?

**Questions:**

- What exactly is meant by "single input UNet" in 4.?
- Why is the UNet the default option here? Wouldn't LSTMs / WaveNet / TCNN would be a better option?

---

> ### Author Response · Authors · 2023-11-21
> **Reply to reviewer gTdv (R3)**
>
> **“Why is the vanilla UNet considered in the first place?”**
>
> **“Why is the UNet the default option here? Wouldn't LSTMs / WaveNet / TCNN would be a better option?”**
>
> We have considered several backbones, focusing on architectures that have demonstrated the highest scores in previous research and have already won ablation studies against alternative models. Our initial choice for a backbone has been a UNet, a selection that aligns with the consensus among imaging diffusion works (Ho et al., 2020; Song et al., 2020a; Dhariwal & Nichol, 2021; Saharia et al., 2022) and has demonstrated effective performance.In addition to the empirical results, Jolicoeur-Martineau et al. (2020) demonstrate that UNets significantly enhance sample quality compared to previous architectures used for denoising score matching. Our second choice of backbone is the Transformer Encoder, used by most motion diffusion works (Kim et al., 2022;Tevet et al., 2023, …). Tevet et al. (2023) conduct ablation studies and show that a Transformer Encoder performs better than a Transformer Decoder and better than a GRU (a GRU is similar to the LSTMs mentioned in R3’s question). In our ablation studies we show that in the case of a single motion input, a UNet performs better than a Transformer.
>
> [Alexia Jolicoeur-Martineau, Rémi Piché-Taillefer, Rémi Tachet des Combes, and Ioan- nis Mitliagkas. Adversarial score matching and improved sampling for image generation. arXiv:2009.05475, 2020.]
>
> **“afaik vanilla UNet [Ronneberger'2015] does not have attention layers, do you imply the unet similar to that used in stable diffusion implementation?“**
>
>  Our "vanilla UNet" is the one employed in early imaging diffusion works, e.g., DDPM (Ho et al., 2020), which includes attention layers. We acknowledge the concern raised by R3 that the term "vanilla UNet" might lead the reader to question whether we are referring to Ronneberger2015's UNet or Ho2020's UNet . We have added a clarification in Section 4 (in blue).
>
> **“What exactly is meant by "single input UNet" in 4?”**
>
> It means a UNet trained on a single input. We revised Section 4 for better clarity. Thank you for bringing it to our attention.

---

### Official Review · Reviewer_qd2e · 2023-11-01

**Soundness:** 3 good
**Presentation:** 4 excellent
**Contribution:** 2 fair
**Rating:** 8
**Confidence:** 3

**Summary:**

The paper introduces SinMDM, a Single Motion Diffusion Model based on the motivation that the number of data for the motion domain is limited. SinMDM learns from a single motion sequence, generating motions true to original motifs, while avoiding overfitting through a lightweight, attention-focused architecture (narrow respective field). It demonstrates various motion tasks, including spatial and temporal composition, as well as Harmonization (style transfer). SinMDM shows high efficiency and good performance.

**Strengths:**

1. The usage of the Diffusion model in the single-motion generation is interesting.

2. Also, this paper demonstrates various interesting applications of motion synthesis with efficiency and good quality.

3. In order to overcome overfitting and promote diversity, SinMDM proposes to use the shallow QnA module to limit the receptive field and relative temporal positional embeddings.

**Weaknesses:**

1. Single-motion generation task and using Diffusion to do single-instance generation is not new. Some concepts have been explored in previous papers. For example, Sinfusion[1] has pointed out that the receptive field needs to be reduced for single-instance generation task. The only difference is that they use ConvNext while SinMDM uses QnA module.

2. Also, for the Harmonization, the way of using guidance injection is from ILVR [2].

Maybe, could you elaborate on this, and emphasize the contribution of this work?

[1] Nikankin, Yaniv, Niv Haim, and Michal Irani. "Sinfusion: Training diffusion models on a single image or video." ICML 2023.

[2] Choi, Jooyoung, et al. "Ilvr: Conditioning method for denoising diffusion probabilistic models."ICCV 2021.

**Questions:**

1. As the receptive field is narrowed, how to guarantee temporal smoothness? As shown in Table 3, using a wide receptive field gives better FID.

2. Comparison with Ganimator in HumanML3D?

---

> ### Author Response · Authors · 2023-11-21
> **Reply to reviewer qd2e (R2)**
>
> **“As the receptive field is narrowed, how to guarantee temporal smoothness? As shown in Table 3, using a wide receptive field gives better FID.”**
>
> Following this question, we added Appendix F (in blue), presenting a study about motion smoothness. In this study, we describe a metric that measures the smoothness fidelity between the generated motion to the ground truth on which the network has been trained.  We show that using a wider receptive field improves smoothness fidelity but degrades the other metrics, thus reducing motion quality. Finally, we introduce a method to enhance smoothness fidelity while maintaining a narrow receptive field, thereby preserving the quality of the results.
>
> As noticed by R2, using a wide receptive field results in better FID. However, a 'near-perfect' FID indicates overfitting to the input motion, as evidenced by poor scores in diversity metrics.
>
> **“Comparison with Ganimator in HumanML3D”**
>
> We added a row to Tab. 2, exhibiting the metric scores of Ganimator on the HumanML3D dataset, and demonstrating once again that SinMDM performs better. Note that Ganimator’s scores demonstrate a very high FID, indicating divergence from the input motion. The updated radar plot reveals a relatively small area for Ganimator, suggesting that it faces challenges in generalizing to other datasets.
> Running Ganimator on HumanmML3D is not straight forward, as Ganimator expects an entirely different data format. To ensure a fair comparison, we transformed the HumanML3D benchmark into the data structure utilized by Ganimator, enabling Ganimator to leverage its skeleton-aware framework. To evaluate the results using the HumanML3D metrics, we reverted them to the HumanML3D format.
>
> **“could you elaborate on this, and emphasize the contribution of this work?”**
>
> Indeed, the performance of SinMDM is built upon the fundamental concepts of Motion Diffusion Models. However, as Motion Diffusion Models typically rely on  large datasets, adapting them to a single motion required a thoughtful redesign. In Section 2, we note that Wang et al. (2022) and Nikankin et al. (2023) use a UNet with limited depth, a strategy not directly applicable to the motion domain. Unlike images with a regularized 2D spatial structure, motion data consists of non-regularized skeletal joints with a temporal axis and fewer degrees of freedom. To bridge this gap, we propose a novel architecture that combines the strengths of the traditional UNET and recent QNA architectures. This effective adaptation to the motion domain enables the application of advanced techniques such as ILVR, as credited in Section 5.2.
>
> We have revised Section 2 to provide an explanation of the distinction between the imaging and motion domains. Thank you for bringing the need for this clarification to our attention.

---

> > ### Author Response · Authors · 2023-11-22
> >
> > Please find a video related to the smoothness study, in the newly uploaded supplementary zip.
> >
> > Description of the smoothness video:
> >
> > At the top right of each slide is the input motion.
> >
> > In the bottom-right corner of each slide are two generated motions. The one on the right has been trained with the smoothness loss (𝜆𝑠𝑚𝑜𝑜𝑡ℎ=1), while the one on the left has been trained without it.
> >
> > Both motions share the same seed, facilitating a temporal similarity in their sub-motions for a more straightforward comparison. When played at normal speed, the motions provide a general impression, but the details are too fast to discern.
> >
> > On the left-hand side of each slide, we showcase a slow-motion sequence spanning several frames, highlighting the enhancement achieved in motions trained with the smoothness loss.
> >
> > The first slide displays an unnatural turn executed by the model trained without the smoothness loss. The second slide depicts a higher-frequency, unnatural leg movement in the model trained without the smoothness loss. The leg changes directions too rapidly, initiates a step, and retracts without completing the step. This leg movement does not align with the motifs of the input motion.

---

> > > ### Comment · Reviewer_qd2e · 2023-11-22
> > >
> > > I appreciate the detailed explanation from the authors. After reading the author's response and the discussion with other reviewers, I will raise my score.

---

### Official Review · Reviewer_ep7Y · 2023-11-01

**Soundness:** 3 good
**Presentation:** 4 excellent
**Contribution:** 2 fair
**Rating:** 6
**Confidence:** 3

**Summary:**

The paper proposes a motion diffusion model (SinMDM) that utilizes a QnA-based UNet architecture in the motion domain. The model incorporates QnA layers, which enable local attention with a temporally narrow receptive field, resulting in improved efficiency in space and time. The use of QnA layers allows for fast and efficient implementation, enhancing the model's performance compared to global attention layers. The paper validates the design choices through experiments and provides a comprehensive list of hyperparameters for reproducibility. The paper demonstrates various applications of single-motion learning using diffusion models. These applications include motion composition, motion harmonization, and straightforward use for long motion generation and crowd animation. The authors showcase the effectiveness of the proposed model in these applications, highlighting its ability to handle complex tasks like harmonization and style transfer.

**Strengths:**

The proposed method is the first work that applies diffusion to single motion learning. The paper is well-written and details are clearly presented. The methodology is well-designed in the way to reduce the range of receptive field to prevent acquiring global context and overfitting. The results in the paper and the supp video are of high quality. Applications are interesting and making use of the diffusion framework. The authors also presents a user study that demonstrates the superiority of the proposed model in terms of diversity, fidelity, and quality.

**Weaknesses:**

- As restricted by the source of data, the results of some editing applications are somehow not that impressive. Similar results can be achieved by simple manipulations without resorting to a powerful diffusion model. E.g. the spatial composition looks no more than cutting-and-pasting the upper/lower body motion, and the style transfer application is like temporally synchronising the pace of the steps of the happy/crouched style video to the content.

- Although a solid work, most techniques used in the methodology are from existing papers, including diffusion for motion, and single motion generation, and the contribution to the community is limited.

**Questions:**

- Are there motion representations other than the adopted one from HumanML3D tested for the proposed method? How do you expect the motion representation could affect the generation performance?

- In what circumstances does the method fail?

---

> ### Author Response · Authors · 2023-11-21
> **Reply to reviewer ep7Y (R1)**
>
> **“Are there motion representations other than the adopted one from HumanML3D tested for the proposed method? How do you expect the motion representation could affect the generation performance?”**
>
> SinMDM is experimented with three different representations: (1) The HumanML3D representation, including joint rotations, global locations and velocities; (2) The Mixamo representation, which includes rotations and global location for the root joint only; and (3) The Zoo representation, which requires extra degrees of freedom and includes rotations + xyz relative location for all joints.
>
> For all datasets, we use rotations of the  6D format, which has been proven by prior work to yield optimal results (See Table 2 in the GANimator paper).
>
> Overall, we showed that our approach is robust with different motion representations.
>
>
> **“In what circumstances does the method fail?”**
>
> In addition to the limitations mentioned in the conclusions section, we observe a failure case that is common to prior works as well - sub-motions are played in an arbitrary order, and this can be a disadvantage when a sequence of sub-motions needs to be executed in a precise order. For example, while doing the bird dance, one should clap hands immediately after shaking the hips. However, single motion algorithms will place these two sub motions in a temporally arbitrary order.
>
> This can be addressed by enlarging the receptive field (at the cost of lower diversity). We added this limitation to the conclusions section (see in blue).
>
> **“Results of some editing applications... Similar results can be achieved by simple manipulations”.**
>
> Many of the applications enabled by SinMDM may also be implemented using designated algorithms. However, these designated algorithms are neither simple nor general, while ours provides a one-stop-shop for all applications, and yields better quality results when compared to simple manipulations.
>
> For spatial composition, using cut-and-paste may yield a motion where the hands and the legs are un-synched, for example, in a walk motion, we expect that when the left hand is at the front, the right hand will also be at the front. In addition, when using cut-and-paste there is no balance compensation when the body leans to a specific direction, for example, when the upper body leans to the right, the pelvis should balance by leaning to the left. The motion generated by cut-and-paste is merely a copy of the upper/lower part of the inputs, hence has no flexibility to adjust them to match each other. On the other hand, the motion generated by SinMDM can be different from the one on which it was trained, thus there can be seamless synchronization between the upper and lower body parts.
> In the context of the style transfer application, while manually synchronizing the pace of steps may yield comparable results, SinMDM achieves this straightforwardly, offering a solution that would otherwise demand a designated, complicated approach, as shown in prior works (Menardais et al., 2004; Lee et al., 2020; Panagiotakis et al., 2013). Temporal synchronization requires the identification of both motions' rhythms, a non-straightforward task by itself. Moreover, stretching one motion according to the rhythm of another is naively done by a  linear interpolation, which can cause motion blur, and may not be linear by nature (say, accelerating a leg motion at the beginning of the step and slowing it as it continues). Our model has no such challenges. It generates the required motion such that it is natural, maintaining the motion motifs of the learned input.
>
> [Stephane Menardais, Richard Kulpa, Franck Multon, and Bruno Arnaldi. Synchronization for dynamic blending of motions. In Proceedings of the 2004 ACM SIGGRAPH/Eurographics symposium on Computer animation, pp. 325–335, 2004.]
>
> [Juyoung Lee, Myungho Lee, Gerard Jounghyun Kim, and Jae-In Hwang. Effects of synchronized leg motion in walk-in-place utilizing deep neural networks for enhanced body ownership and sense of presence in vr. In Proceedings of the 26th ACM Symposium on Virtual Reality Software and Technology, pp. 1–10, 2020.]
>
> [Costas Panagiotakis, Andre Holzapfel, Damien Michel, and Antonis A Argyros. Beat synchronous dance animation based on visual analysis of human motion and audio analysis of music tempo. In International symposium on visual computing, pp. 118–127. Springer, 2013.]

---

### Author Response · Authors · 2023-11-21
**General Reply**

We thank the reviewers for the thorough review and thoughtful comments. We are pleased to see that reviewers think SinMDM has a well-designed methodology (ep7Y, qd2e, gTdv) with high-quality results (ep7Y, qd2e, gTdv), thorough numerical evaluations with state-of-the-art results (gTdv, BL7j) and useful applications (ep7Y, qd2e, BL7j). We are also happy to see that the reviewers recognize SinMDM’s high efficiency (qd2e), reproducible technique (BL7j), and clear presentation (ep7Y, gTdv, BL7j).

Our work is based on Motion Diffusion Models, moving away from the requirement of extensive datasets to showcasing their capabilities with just a single motion. Training on a single motion is particularly essential in the motion domain, where the number of data instances is limited, particularly for the animation of non-humans.

The adaptation to single instances prompted a redesign to prevent overfitting. This was achieved by employing a shallow network with local attention layers, narrowing the receptive field and encouraging motion diversity. This redesign has enabled zero-shot applications, and has yielded state-of-the-art results both quantitatively and qualitatively, surpassing previous single motion generation works by a significant margin.

In the updated version of the paper, changes are marked in blue. The following summarizes the major parts added to the paper based on the reviewer’s suggestions:
1.  A Comparison with Ganimator on HumanML3D
2. Appendix F: A study about motion smoothness, its correlation to receptive field width, a metric to measure it and ways to ameliorate it.

We deeply appreciate your reviews and feel that the new revision brought is more complete thanks to them. In addition, we address individual comments and questions by commenting on your reviews.

---

### Meta-Review · Area_Chair_BhyJ · 2023-12-10

**Metareview:**

The paper received favorable reviews from the outset. The reviewers highlighted that the proposed idea of using a diffusion model for single motion generation is sounds and interesting and the results are impressive. During the discussion period the authors did a good job in providing further details about their approached, such that the scores went up. The AC concurs and decides to accept the paper. Congrats!

**Justification For Why Not Higher Score:**

This is an interesting manuscript, however, the AC doesn't think it's fundamental enough to secure an oral. Besides, the AC is uncertain if the topic of the manuscript is interesting for a broader community rather than a niche area of researchers focusing on animation.

**Justification For Why Not Lower Score:**

Due to the niche domain the paper can be downgraded to a poster acceptance

---

### Decision · Program_Chairs · 2024-01-16

Accept (spotlight)